# Fenretinide in Cancer and Neurological Disease: A Two-Face Janus Molecule

**DOI:** 10.3390/ijms23137426

**Published:** 2022-07-04

**Authors:** Rosa Luisa Potenza, Pietro Lodeserto, Isabella Orienti

**Affiliations:** 1National Center for Drug Research and Evaluation, Istituto Superiore di Sanità, 00161 Rome, Italy; 2Department of Pharmacy and Biotechnology, Alma Mater Studiorum-University of Bologna, 40127 Bologna, Italy; pietro.lodeserto@unibo.it (P.L.); isabella.orienti@unibo.it (I.O.)

**Keywords:** fenretinide, anticancer drugs, nanomicellar formulations, repositioning, neuroinflammation, oxidative stress, neuroprotection, hormesis

## Abstract

Recently, several chemotherapeutic drugs have been repositioned in neurological diseases, based on common biological backgrounds and the inverse comorbidity between cancer and neurodegenerative diseases. Fenretinide (all-trans-N-(4-hydroxyphenyl) retinamide, 4-HPR) is a synthetic derivative of all-trans-retinoic acid initially proposed in anticancer therapy for its antitumor effects combined with limited toxicity. Subsequently, fenretinide has been proposed for other diseases, for which it was not intentionally designed for, due to its ability to influence different biological pathways, providing a broad spectrum of pharmacological effects. Here, we review the most relevant preclinical and clinical findings from fenretinide and discuss its therapeutic role towards cancer and neurological diseases, highlighting the hormetic behavior of this pleiotropic molecule.

## 1. Introduction

Cancer and neurodegeneration share leadership as causes of morbidity and death worldwide. They can be thought as disease mechanisms at opposite ends: while in neurodegeneration, induction of inflammatory genes and suppression of cell-cycle genes are the prominent signals; the opposite happens in cancer.

In both cancer and neurodegeneration, apoptosis, autophagy and necrosis interact with each other to establish the cell fate with an ultimate aim of favoring survival and proliferation in cancer and rescuing the healthier unaffected cells in neurodegeneration [1].

In cancer cells, inhibition of apoptosis, by the increase of anti-apoptotic proteins and decrease of pro-apoptotic molecules, is the main mechanism promoting cell survival and proliferation. In addition, autophagy further contributes to cell survival in response to stress conditions like hypoxia, and promotes chemoresistance to chemotherapy [2].

In neurodegeneration, on the contrary, autophagy is the main mechanism to sustain cell survival and oppose cell death. However, as deterioration increases to levels beyond which repair mechanisms prove ineffective, cells block autophagy and initiate the apoptotic cascades [3].

Therefore, the mechanisms regulating cell death and survival operate in opposite directions in cancer and neurodegeneration. This could partially explain the inverse comorbidity between cancer and neurodegenerative diseases, as reported in Parkinson disease (PD) [4] and in Alzheimer disease (AD) [5].

In addition, the therapeutic modalities operate in opposite directions in these two pathologies. Antitumor therapies are aimed at triggering apoptosis to inhibit cell proliferation, tumor mass increase and tumors spreading throughout the body. In neurodegenerative diseases, instead, therapies rely on inhibition of apoptosis and the promotion of autophagy, which represents the main rescue mechanism against cell death. Indeed, autophagy is fundamental in neurons to preserve their physiological balance [6]. Failure of autophagy easily leads to neurodegeneration and cell death [7].

The different responses of tumor cells and neurons to drugs affecting apoptotic and autophagy pathways suggest the repositioning of anticancer drugs for neuroprotection, or the repositioning in the opposite direction, to develop novel therapeutics options [8].

Chemotherapy drugs such as kinase inhibitors, antimetabolites, alkylating agents and antibodies have been repurposed in neurodegenerative diseases based on shared target genes in cancer and neurodegeneration [8]. However, their toxicity and induction of drug resistance are the main drawbacks of their repeated use in both diseases.

Among the molecules capable of regulating autophagy and apoptosis, all-trans-N-(4-hydroxyphenyl) retinamide (4-HPR or fenretinide) is particularly interesting.

Fenretinide is a synthetic derivative of all-trans-retinoic acid (ATRA), produced in the USA in the 1960s [9] and initially proposed as an anticancer agent due to its high antitumor activity, favorable toxicological profile and lack of induction of resistance. Currently, it is also studied in many other diseases for its ability to influence several biological pathways and provide a broad spectrum of pharmacological effects.

Fenretinide behaves as an atypical retinoid with both retinoid acid receptor-dependent and independent activities, showing a different pharmacological behavior compared to retinoic acid [10].

Indeed, unlike retinoic acid, fenretinide inhibits cell growth through apoptosis rather than differentiation and its apoptotic effect involves generation of reactive oxygen species (ROS) and lipid second messengers [11] that are retinoid acid receptor-independent events [12].

Furthermore, fenretinide binds with high affinity the specific retinol-binding protein 4 (RBP4) instead of retinol, thus inhibiting the formation of the retinol complex with RBP4 and transthyretin (TTR), which is physiologically released from the liver to the bloodstream with the function of distributing retinol to body tissues via specific membrane receptors. This way, fenretinide, unlike most retinoids, reduces the circulating levels of retinol by inhibiting the formation of the retinol/RBP4/TTR complex and increases the renal clearance of RBP4 by inhibiting its interaction with TTR [13].

High serum RBP4 concentrations have been associated with dysregulation of energy metabolism, insulin resistance, diabetes mellitus and obesity [14]. Therefore, due to its ability to decrease RBP4 serum levels, fenretinide has been evaluated to prevent insulin resistance and glucose tolerance in obese mice and in overweight humans [15,16].

Fenretinide modulation of insulin resistance has also been attributed to the inhibition of dihydroceramide desaturase (DES1), an enzyme engaged in de novo biosynthesis of ceramides which antagonizes insulin action. DES1 inhibition obtained by fenretinide treatment has been shown to block lipid-induced insulin resistance through depletion of cell ceramides (Cer) and accumulation of precursor dihydroceramides (DhCer) [17]. In addition, DES1 inhibition induced autophagy and decreased cancer cell growth [18].

DhCer and Cer may have several functions in cellular processes and sometimes antagonistic effects [19]. Their plasma concentrations have been related to metabolic diseases, thus representing key players and potential biomarkers in pathologies ranging from diabetes to cancer and neurodegenerative diseases. DhCer levels, in particular, directly influence the lipid compositions of organelle membranes and any change can induce stress responses that mediate their effects on cell death or survival [18].

## 2. Fenretinide in Cancer

Typical retinoids induce antitumor effects mainly by decreasing the expression of anti-apoptotic bcl-2 family genes binding retinoic acid receptors (RAR) and retinoid X receptors (RXR): RXR-alpha, RXR-beta and RXR-gamma, which are ligand-activated transcription factors regulating several cellular processes including growth, differentiation and apoptosis [20].

The antitumor effect of fenretinide is more complex and encompasses at least four different mechanisms of action (Figure 1).

Firstly, fenretinide selectively binds the RARβ receptors, increasing their expression and inducing translocation of the nuclear receptor Nur77, as a dimer with RARβ, from the nucleus to the cytoplasm. In the cytoplasm, Nur77 binds Bcl-2, allowing their conformational change towards pro-apoptotic structures that expose the BH3 domain [21,22,23,24].

In human neuroblastoma cells, fenretinide induced apoptosis by a caspase-dependent mechanism. Cell treatment with RARβ/γ antagonists inhibited apoptosis, while inhibitors of RARα had no effect [25].

The second mechanism is DES1 inhibition, which produces an increased DhCer/Cer ratio in cell membranes. This causes endoplasmic reticulum stress and a consequent block of the phosphatidylinositol-3-kinase (PI3K)/Akt/mammalian target of rapamycin (mTOR) and NF-kB signaling pathways. In addition, the endoplasmic reticulum (ER) stress induces phosphorylation of the a-subunit of eukaryotic initiation factor 2 (eIF2) due to the protein kinase RNA-like endoplasmic reticulum kinase (PERK), which therefore can no longer carry out its activity as an initiator of protein translation [18].

In cancer cells, fenretinide increased DhCer levels in a dose-dependent manner, whereas no significant changes were seen in Cers content [26]. In prostate cancer cells exposed to fenretinide, DhCers promoted autophagy by inducing the formation of autophagosomes [27]; although autophagy can act as a pro-survival mechanism, it can also promote cell death and excessive autophagic activity can lead to digestion of the entire cell [28].

The third mechanism is the inhibition of mTOR. The molecular structure of fenretinide matches the binding site of mTOR with ATP, suppressing the activities of both the mTORC1 and the mTORC2 complexes and the related PI3K/AKT pathway; as a consequence, a proliferation decrease is obtained by inhibition of the PI3K/AKT pathway and the translational regulation activity of mTORC2. In vitro studies confirmed that fenretinide inhibits non-small-cell lung cancer cells growth attenuating mTOR downstream signaling. Moreover, knockdown of mTOR in cancer cells decreased their sensitivity to fenretinide [29].

The fourth mechanism is based on the ability of fenretinide to replace retinol in the mitochondrial signalosome, composed of a signal adaptor protein (p66Shc), protein kinase Cδ (PKCδ) and cytochrome c (CytC). The retinol/p66Shc-PKCδ-CytC signalosome behaves as a catalyst for the passage of electrons from PKCδ to CytC and vice versa. The electron donated by PKCδ to cytochrome oxidizes the zinc-finger domain of the kinase, resulting in a structural modification which exposes the binding site to the substrate and the binding site of ATP. This activation induces, in turn, the activation of pyruvate dehydrogenase kinase 2 (PDK2) through phosphorylation, which allows pyruvate dehydrogenase phosphatase 1 (PDP1) and 2 (PDP2) to activate the pyruvate dehydrogenase complex (PDHC) that feeds the tricarboxylic acid cycle (TCA). This makes NADH available for the oxidative phosphorylation (OXPHOS) where the electron transport chain generates ROS and ATP.

The electron flow generated by this mechanism is reversible only in the presence of physiological concentrations of retinol that allow the signalosome to switch from activation to inactivation depending on the energy demand of the cell; on the contrary, when retinol is replaced by fenretinide or retinoic acid, the mechanism is not reversible anymore and PDHC is locked in an activation state leading to ROS overproduction (Figure 1).

Cellular ROS increase is one of the triggering signals for the intrinsic pathway of apoptosis, leading to the increase in mitochondrial membrane permeability and release of pro-apoptotic proteins in the cytoplasm (such as CytC and the direct inhibitor of apoptosis-binding protein with low pI DIABLO) leading to cell death via caspase activation [30].

In tumor cells, apoptosis is activated only in the presence of high ROS levels, while low levels activate autophagy. This depends on DJ-1, a redox sensor protein that detects ROS concentration and determines the cellular response. Under moderate ROS increase, the mild oxidized form of DJ-1 links and inactivates the apoptosis signal-regulating kinase 1 (ASK1) in the *p*-38 apoptotic pathway, thus blocking apoptosis, but it does not interfere with ASK1 in the c-Jun N-terminal kinase (JNK) autophagy pathway; in this way, apoptosis is suppressed and cell viability is maintained by autophagy. On the contrary, in the presence of high ROS levels, perceived as lethal by the cell, an excessive oxidation of DJ-1 induces its dissociation from ASK1 with consequent activation of the *p*38 apoptotic pathway [31] involving CytC release, caspase-9 activation and apoptosis. The fine regulation of cell response to ROS increase is responsible for the different effect of fenretinide in tumors vs normal cells and in different tumor types.

Indeed, a cell response such as autophagy (causing cell survival) or apoptosis (causing cell death) is strictly dependent on the threshold of cell tolerance towards ROS increase that changes in different cell types and conditions. However, as a rule, high fenretinide concentrations are expected to promote apoptosis by induction of high ROS levels, that activate the *p*-38 apoptotic pathway and, by inhibition of PI3K/AKT/mTORC, DES1 and RARβ-Nur77 pathways that block proliferation and further stimulate apoptosis.

Conversely, low fenretinide concentrations are expected to favor cell survival over cell death due to the moderate increase of ROS production, which activates the JNK autophagy pathway. In addition, low fenretinide concentrations are not expected to inhibit the PI3K/AKT/mTORC, DES1 and RARβ-Nur77 proliferation and apoptosis-controlling pathways. This relationship between concentration and type of response has been observed in glioma cell lines in which fenretinide induced apoptosis at concentrations of 10 μM and autophagy at 5 μM; at concentrations higher than 10 μM, the glioma cells undergoing apoptosis also showed characteristic features of autophagy [32].

## 3. Clinical and Preclinical Evaluation of Fenretinide in Cancer

The activity of fenretinide has been demonstrated in vitro in a wide range of cancer cells. A minimum concentration of 10 µM has been required to provide 50–90% cell growth inhibition and apoptosis after 72 h of exposure [33].

Clinical trials of fenretinide to date have shown high variability in results, indicating that the minimum active concentration is not always achieved in the tumor mass.

As is well-known, the achievement of an active drug concentration in solid tumors depends mainly on the permeability of the tumor to drug penetration and the drug plasma concentration that controls blood-tumor drug distribution.

Low aqueous solubility has often been a major obstacle to the development and clinical use of antitumor drugs, as their administration by any route has frequently failed to provide adequate drug plasma concentrations for activity.

Fenretinide is a molecule characterized by very low aqueous solubility and high lipophilicity, which severely limits its bioavailability and provides low plasma concentrations that are often insufficient to allow drug penetration into the solid tumor at levels required for activity.

For drugs with low aqueous solubility, such as fenretinide, the formulation may become a means of increasing drug solubilization in plasma and body fluids and/or controlling the drug distribution among different body compartments, thus improving overall activity.

Clinical trials conducted so far on fenretinide have mainly used oral soft gelatin capsules consisting of a fenretinide dispersion in corn oil and polysorbate 80, which have been the most widely used formulations in clinical trials so far.

In a phase I trial, the capsules were administered to children with neuroblastoma, in a range of doses from 100 to 4000 mg/m^2^ (single daily dose) for 28 days followed by a seven-day drug holiday. After the first cycle (day 28), fenretinide peak plasma concentrations ranged from 1.3 to 12.9 µM in a dose-dependent manner; no complete or partial responses were observed, but 77% of the patients experienced stable disease for a median of 23 months [34]. A pharmacokinetic study with the fenretinide capsules, at the same doses and schedule, reported average plasma drug concentrations of 9.9 µM at the highest dose [35].

Doses from 350 to 3300 mg/m^2^ (divided twice or thrice daily administrations) for days 1 to 7 every three weeks provided 1 complete response and 13 stable disease responses out of 30 evaluable neuroblastoma patients who completed at least eight cycles. The maximum tolerated dose (2475 mg/m^2^) provided a fenretinide plasma concentration of 9.9 µM at the end of the first cycle [36].

In a phase II study, patients with renal cell carcinoma were treated with 1800 mg/m^2^ (divided twice daily) of fenretinide capsules and minimal activity was observed in 18 evaluable patients. The intratumor concentrations of fenretinide, in tumor biopsies from four patients, were within 3.6–7.9 µM, thus under the 10 µM threshold for activity [37].

The same dosage and schedule with fenretinide capsules failed to produce objective responses in patients with small-cell lung cancer (for 7 days every three weeks, with a mean plasma concentration of fenretinide of 7.4 µM on day 7 of cycle 1) [38], whereas patients showed limited activity against prostate cancer administered at 900 mg/m^2^ (twice daily) [39,40].

The capsule formulation failed to show beneficial effects in patients with recurrent malignant glioma [41] and ascitic ovarian cancer [42]; thus, trials were stopped due to lack of efficacy. In an ovarian cancer phase II study, 42% of patients treated with fenretinide capsules (1800 mg/m^2^ twice daily) showed stable disease for a median duration of 7.2 months with fenretinide plasma concentrations ranging from 3.1 to 12.5 µM and no objective responses. However, progression-free survival and overall survival improved in patients with the highest fenretinide concentration levels [43].

Again, in a phase II study, 31 patients with breast cancer or melanoma, treated daily (from 10 to 300 days) with fenretinide capsules, showed partial responses [44].

The above-mentioned studies showed that the highest average plasma concentration of 9.9 µM was obtained by repeated administration of the fenretinide capsules at the drug dose of 2475 mg/m^2^ for 7 days. This value approached 10 µM, the minimum concentration providing activity in in vitro studies.

Other fenretinide formulations were able to raise the drug plasma concentrations over the values achieved with the gelatin capsules, generating superior clinical responses.

A formulation based on fenretinide mixed with a lipid matrix called LYM-X-SORB™ and dispersed in a liquid nutritional drink was evaluated in a phase I study in patients with neuroblastoma at doses ranging from 352 to 2210 mg/m^2^ (divided twice daily) for seven days every three weeks. Out of 29 evaluable patients, 4 had complete responses and 6 had stable disease; fenretinide mean peak plasma levels reached 21 µM at 1700 mg/m^2^/day on day 6 of cycle 1 [45].

An intravenous lipid emulsion containing fenretinide (infused in continuo at 905–1414 mg/m^2^/day for 5 days in 21-day cycles) was administered in 23 patients with advanced solid tumors enrolled in a phase I study; fenretinide plasma steady-state concentrations were in the range 17.87 µM–38 µM and although no patients had objective responses, five patients showed stable disease as best response [46].

More recently, novel nanoformulations were prepared to improve fenretinide bioavailability. They were based on phospholipid–liquid triglyceride mixtures with or without cyclodextrins. In an aqueous environment, they were able to self-assemble with formation of nanomicelle-like structures that entrapped fenretinide, thus improving the drug’s aqueous solubilization. Administered by oral or intravenous routes, these nanoformulations strongly increased fenretinide plasma concentrations and provided significant antitumor activity in preclinical studies.

Among them, a nanoformulation, called bionanofenretinide (Bio-nFeR), provided drug plasma concentrations and tumor concentrations tenfold higher than those obtained with the gelatin capsules at the same administration dose. It was obtained by drug encapsulation in an ion-pair stabilized lipid matrix. Administered by oral route in animal models of human lung cancer, colon cancer and melanoma at doses 10–200 mg/kg, Bio-nFeR significantly decreased tumor growth. A single treatment at 100 mg/kg provided 9.2 µM drug plasma concentration and 9.6 µM drug concentration in tumors. The gelatin capsules at the same administered dose provided 1.0 µM plasma concentration and 1.5 µM tumor concentration [47]. Another nanoformulation, obtained by phospholipid–liquid triglyceride and cyclodextrins, was administered, in combination with lenalidomide, by intravenous injection in animal models of human neuroblastoma at 30 mg/kg fenretinide three times/week for 3 weeks. The treatment provided 100% survival at the end of the experiment with 10.06 µM plasma concentrations and 54.47 µM tumor concentrations [48,49].

The same nanoformulation also showed significant therapeutic efficacy in brain tumor models (DIPG-07 and RARE-08), indicating its ability to cross the blood–brain barrier (BBB) [50,51].

## 4. Fenretinide in Neurological Diseases

Due to its high metabolic rate and relatively low levels of antioxidant defenses, the brain is highly sensitive to oxidative stress and, as a consequence, neurological diseases are often characterized by increased oxidative stress and reduced ability of the antioxidant system to counter free radicals [52,53]. Other specific features of neurological diseases are increased lipid peroxidation, DNA and protein oxidation [54,55].

In the central nervous system (CNS), ROS increase generates dysregulation of the inflammatory response through the activation of microglia and astrocytes and subsequent release of cytokines and inflammatory mediators that can disrupt the blood–brain barrier’s integrity; this allows leukocytes, such as T cells and macrophages, to infiltrate the CNS and further increase neuroinflammation [56,57].

The ability of fenretinide to modulate two significant features of neurodegeneration (oxidative stress and inflammatory response), combined with its high lipophilic character that allows, in suitable formulations, to cross the blood–brain barrier [41,58], suggested to several authors its potential use in treating pathological conditions of the nervous system.

The anti-inflammatory effect of fenretinide has been attributed to its ability to downregulate the production of arachidonic acid (AA), a proinflammatory omega-6 polyunsaturated fatty acid, and to increase the levels of the anti-inflammatory omega-3 polyunsaturated docosahexaenoic acid (DHA) in macrophages [59,60,61]. In fact, DHA was shown to act as an inhibitor of various toll-like receptors (TLRs) and TLR-mediated signaling pathways, including ERK1/2 [62]. Fenretinide inhibition of the ERK1/2 pathway decreased inflammatory mediators’ genes expression, including TNF, IL-6, CCL2 and CCL5, in infected or in lipopolysaccharide endotoxin (LPS)-stimulated macrophages without generation of a widespread shutdown of signaling mechanisms [63,64].

Furthermore, fenretinide was shown to decrease proinflammatory cytokine secretion by enhancing the expression of the peroxisome proliferator-activated receptor gamma (PPARγ) in macrophages with consequent inhibition of inflammatory programs of gene expression (Figure 2). Fenretinide not only increased the expression of PPARγ but also acted as a ligand for PPARγ, thus increasing its activity [65].

The ability of fenretinide to decrease oxidative stress relies on improved expression of the nuclear factor erythroid 2-related factor 2 (Nrf2) and its nuclear translocation that promote the transcription of Nrf2-antioxidant responsive element (ARE), as demonstrated in mouse brain endothelial cell line [66].

Moreover, the activation of autophagy, which can take place in both tumor and normal cells at subtoxic fenretinide doses (Figure 2), can represent a positive effect in neurodegeneration as autophagy, at basal levels, is a physiologic mechanism to sustain cell survival and oppose cell death.

Finally, due to its ability to decrease RBP4 serum levels and, consequentially, to modulate insulin resistance and glucose intolerance, fenretinide could have a positive impact on neurological diseases in which glucose metabolic dysfunctions represent a risk factor [67].

## 5. Preclinical Evaluation of Fenretinide in Neurological Disease

### 5.1. Multiple Sclerosis

In experimental allergic encephalomyelitis (EAE) mice, low doses (3 mg/kg/day) of oral fenretinide, very far from those required to produce cytotoxic effects in tumor cells [68], were able to reduce severity of EAE symptoms [69]. Experimental autoimmune encephalomyelitis (EAE), in which susceptible mice strains are immunized with myelin basic protein (MBP), is one of the common models for relapsing/remitting multiple sclerosis (MS), an (auto)immune-driven neurological disease specifically affecting the central nervous system. EAE mice mirror some important histopathological and immunological hallmarks of human MS characterized by infiltration of immune cells into CNS and demyelination. The disease course induced by MBP, characterized by an acute paralytic episode and a significant amount of Wallerian degeneration in the spinal cord, was counteracted by dietary administration of fenretinide. More importantly, fenretinide reduced inflammation and demyelination and lessened the severity of neurological deficits in EAE mice even when administered after disease onset. Furthermore, fenretinide stimulated T cell differentiation towards Th2 type with increased release of IL-4 [68]. This condition had already been shown to be beneficial in the EAE model [70] since T cell-induced microglia switch toward inflammatory phenotype is crucial in MS.

Fenretinide also reduced the expression of proinflammatory cytokines in LPS-stimulated macrophages [63]; this appears to be very relevant considering that macrophages dominate sites of CNS injury in which they can promote both injury (“classically activated” M1 macrophages) and repair (“alternatively activated” anti-inflammatory (M2) cells) mechanisms [71].

### 5.2. Spinal Cord Injury

In activated macrophages (and microglia), different stimuli lead to alterations in their phospholipid-bound fatty acid metabolism, resulting in secretion of several inflammatory lipid mediators; lipid metabolism, *vice versa*, significantly affects macrophage phenotype and functions [72] and plays a critical role in the activation of both M1 and M2 macrophages [73].

Lipids represents about 50% of brain dry weight, a very high concentration that in the human body is second only to adipose tissue [74]. Lipid dysmetabolism, leading to accumulation of fatty acid metabolites, impairs central and peripheral nervous system function in humans.

The omega-3 (DHA) and omega-6 (AA) polyunsaturated fatty acids (PUFA), major constituents of cell membranes, are the most susceptible to lipid peroxidation of all fatty acids. AA and the DHA act as potent signaling molecules that promote or dampen, respectively, molecular pathways involved in the inflammatory process [75]. Furthermore, AA and DHA are precursors of large families of other regulatory molecules, including AA-derived prostanoids (prostaglandins and leukotrienes), and the docosanoid resolvins and neuroprotectins, able to amplify the pro- or anti-inflammatory signaling cascade, respectively [76].

Fenretinide, as already mentioned, downregulates the activation of the ERK1/2 mitogen-activated protein (MAP) kinase pathway and the expression of inflammatory mediators via normalization of the AA/DHA balance [63,64]. Thus, fenretinide can support the health of the central nervous system due to its ability to modulate fatty acid metabolism and lipid mediators.

This mechanism has been called into question in reducing expression of proinflammatory genes and decreasing oxidative stress in a mice model of spinal cord injury, in which fenretinide beneficially acted on motor deficit. Oral administration of low daily doses of fenretinide (5 mg/kg/die) modulated PUFA levels in plasma and in injured CNS tissue of mice, after spinal cord contusion. A significant improvement in locomotor recovery was also recorded [77].

### 5.3. Amyotrophic Lateral Sclerosis

The unpublished doctoral thesis by T. Skinner [78], an author of the paper of Lopez-Vales et al. [77], describes a beneficial role of the same doses of fenretinide in a faster progressive mice model of amyotrophic lateral sclerosis (ALS), a fatal neurodegenerative disease characterized by selective death of motor neurons causing progressive muscle atrophy and spasticity. In ALS, reduced mitochondrial function is related to increased mitochondrial protein and lipid oxidation [79]; thus, oxidative stress and inflammation may be present at the same time both in the motor neurons and in neighboring cell populations.

In the spinal cords of ALS mice chronically treated (from presyptomatic stage of the disease) with fenretinide at low oral doses (5 mg/kg), Skinner observed increased DHA/AA ratio, decreased lipid peroxidation and gliosis (the hallmarks of neuroinflammation in ALS) and a significant reduction in the expression of TNF-α and iNOS proinflammatory mediators. These histopathologic findings correlated well with an improvement in motor functions and increased survival time in affected female mice.

Very recently, we demonstrated that low doses (10 mg/kg) of a new nanomicellar fenretinide formulation, administered by intraperitoneal route, were able to extend the survival of female, but not male, ALS mice, even when administered after the onset of motor symptoms [80]. This is an important translational item, as starting animal treatment before the symptom onset is considered a methodological weakness for the predictive values of preclinical research with ALS mice [81]. We also demonstrated that expression of ALS-linked SOD1 mutation in cultured motoneuron cells resulted in mitochondrial dysfunction, which can be reversed by treatment with fenretinide [80].

### 5.4. Alzheimer Disease

Retinoids can act upon multiple Alzheimer disease-associated targets, a neurodegenerative disease currently ranked as the most common cause of dementia among older adults [82]. The main histopathological hallmark of AD is the aggregation of Aβ peptides which accumulate into senile and neuritic plaques in different areas of the brain. Age is the primary risk factor of AD; however, it is still unclear whether AD occurs in old age or if it originates earlier. Retinoid deficiency is greatest in late-onset Alzheimer disease (LOAD) [83] which accounts for 90% of AD cases [84].

Accordingly, in a mouse model of AD, it was shown that administration of all-trans retinoic acid recovered adult neurogenesis in the hippocampus through inhibition of microglial activation [85].

Other important risk factors in LOAD are represented by glucose intolerance and insulin resistance, which are mediators of AD neurodegeneration [86], highlighting the impact of metabolic factors in premature cognitive aging and risk of developing dementia.

In neuron-specific human BACE1 knock-in mice, an animal model of sporadic AD, fenretinide (0.04% *w*/*w* in diet, approximately corresponding to 40–50 mg/kg/die) has been shown to reduce adiposity gain and serum leptin, a key regulator hormone in the control of body weight, metabolism and glucose homeostasis [87].

In the same animal model, Plucińska and collaborators [88] demonstrated the ability of fenretinide (chronically administered with the diet) to decrease ER stress and neuroinflammation while boosting the expression of sAPPα, the neuroprotective fragment of the amyloid precursor protein. In these mice, fenretinide significantly prevented early signs of spatial memory deficits, restored the cerebral expression of full-length APP and lowered the accumulation of toxic Aβ oligomers. However, fenretinide effects on AD phenotype appeared independent of its ability to modulate the systemic glucose homeostasis in BACE1 mice, suggesting that fenretinide may affect β-site cleavage of APP via an indirect mechanism.

### 5.5. Depression

Fenretinide activity was also proved in an animal model of lipopolysaccharide (LPS)-induced brain injury. In vivo, LPS endotoxin administration provided immune cell activation, proinflammatory cytokine release, disruption of the BBB integrity and increased oxidative damage [89], leading to the damage of brain tissues. LPS administration is widely used to induce depressive-like behavior in rodents as LPS-treated mice mimic human depressive symptoms with an earlier “sickness behavior” phase which includes fever, anorexia, reduction of locomotion and a decrease in social interaction, followed by several depressive-like behaviors [90].

LPS administration elevates the immobility time that reflects a measure of behavioral despair when placing a rodent in an uncomfortable situation from which escape is impossible; interestingly, intraperitoneal injections of fenretinide (20 and 40 mg/kg) reverted this depressive behavior. Fenretinide also reduced BBB dysfunction in LPS-induced brain injury and promoted Nrf2-dependent antioxidative signaling [66].

The Nrf2 pathway is naturally activated as an endogenous compensatory mechanism [91], and treatment with compounds able to cross the BBB and activate the Nrf2 pathway have been indicated as emerging therapeutic strategies against the oxidative stress damage in the central nervous system, supporting the neuroprotective effect of fenretinide which exhibits these abilities [66].

This may also explain the different outcome observed in rats and mice chronically treated with ATRA that, contrary to fenretinide, induce depression-related behaviors [92,93]. As reported by Wang and collaborators, ATRA acts as an inhibitor of transcription factor Nrf2 through the binding of the retinoic acid receptor alpha [94] whereas fenretinide promotes Nrf2-antioxidant responsive element (ARE) transcription activity to exert its antioxidant function [66,95].

Fenretinide can indirectly modulate antioxidant Nrf2 activity increasing β-carotene accumulation [96], which was shown to prevent neuronal loss in ROS-associated brain diseases also by Nrf2 increase [97,98]. In a phase III clinical trial (ClinicalTrials.gov Identifier: NCT00534898), fenretinide (5 mg/kg/die) was proposed as add-on to stable ongoing antipsychotic treatment in patients suffering from schizophrenia. Although the study was withdrawn due to lack of financial support, the rationale behind the proposal (the retinoid dysregulation hypothesis) suggests that fenretinide should be considered and evaluated as a treatment in other psychiatric animal models other than depression.

### 5.6. Retinopathies

Despite its peripheral location, the retina, the neural portion of the eye, represents an extension of the central nervous system, with specialized immune responses similar to those of the brain and spinal cord. Like the CNS, the retina is rich in lipid membranes and subject to a high metabolism making it particularly susceptible to oxidative stress [99].

The visual effects induced by fenretinide treatment appeared to be related to a reduced availability of circulating retinol [96]. The first observations emerged from the adverse effects reported in clinical trials employing high doses of fenretinide (600–800 mg/day) showing significant electroretinogram alterations and impaired dark adaptation [100]. Subsequently, Decensi and collaborators reported that the daily dose of 200 mg, which is currently used in cancer chemoprevention trials, exerted only slight changes [101]. Again, clinical data seemed to suggest that fenretinide exerted a dose-dependent degree of retinal function inhibition.

Based on its ability to reduce circulating levels of retinol, fenretinide treatment has been proposed in geographic atrophy, an advanced form of age-related macular degeneration affecting the retina characterized by an excessive accumulation of retinol-based toxins. Unlike other organs, the uptake of retinol by the eye is largely dependent on delivery by the RBP complex which can be efficiently modulated by fenretinide through a rapid elimination of the complex via urine. In a 2-year placebo-controlled double-masked trial, 100 and 300 mg of fenretinide were orally administered every day in 246 geographic atrophy patients [102]. As expected, fenretinide administration induced a dose-dependent reversible reduction in serum RBP-retinol with visual disturbance and night blindness as the most common adverse effects in patients treated with the higher doses. More importantly, limiting retinol uptake from the serum into retinal pigment epithelium has been associated with a decrease in lesion growth rates, supporting the potential of fenretinide in the management of retinopathy.

Based on the study in geographic atrophy patients, fenretinide was also proposed as a potential treatment in the Stargardt’s macular degeneration disease [103], a genetic eye disorder that causes progressive vision loss; to date, however, no clinical or preclinical data are available.

## 6. Conclusions and Future Perspectives

From our review clearly emerges that fenretinide may have multiple clinical applications depending on dosage/concentration as well as on the different mechanisms underlying the specific pathological condition (Table 1).

As already reported, high doses of fenretinide can activate acute responses leading to ROS increase and cell death [31], while, at subtoxic concentrations, fenretinide can stimulate adaptive stress responses [30]. Specifically, at lethal levels of oxidative stress, induced by high fenretinide concentrations, excessive oxidized DJ-1 (an oxidative stress response protein) induces *p*38 activation enabling the cells to commit to apoptosis. On the contrary, in the presence of mild oxidative stress, as provided by low fenretinide concentrations, mildly oxidized DJ-1 is recruited to inhibit the activity of ASK1 thus inhibiting apoptosis, maintaining cell viability by autophagy activation [31] and providing an overall antioxidant action [104].

Thus, fenretinide behavior appears consistent with a hormetic-biphasic dose response (Figure 3).

As it is well-known, hormetic behaviors are physiological biphasic responses of cells that control adaptation to stress. Low doses of stressors (e.g., toxins) provide benefit to the cells while high doses induce toxic responses. This behavior, if properly exploited, could have potential therapeutic applications, as already discussed for cancer [105] and neurodegenerative diseases [106].

The hormetic behavior of fenretinide, already reported for other carotenoids [107,108], is demonstrated by the different drug doses necessary to provide efficacy in tumors or neurological disease, respectively.

In vitro, the neuroprotective effect of fenretinide is present at concentrations ≤ 0.1 µM [80], whereas concentrations of at least 10 µM fenretinide are required to induce apoptosis in most cancer cells [33].

In vivo, as described in the previous sections, antitumor efficacy of fenretinide was reported at doses ranging from 50–500 mg/kg by oral route and 24–30 mg/kg by intravenous administration, while efficacy in neurological diseases was observed at much lower doses, ranging from 3–40 mg/kg either by oral or intraperitoneal routes.

Therefore, the activity of fenretinide as pro-apoptotic/antitumor or pro-autophagic/adjuvant strictly depends on its concentration in the pathological site (tumor or CNS tissues). This concentration is related to the dose, the route of administration and the type of formulation.

The type of formulation has proved particularly important with fenretinide, as this drug is characterized by extremely low aqueous solubility and hydrophobicity that strongly limit its availability in body fluids thus preventing its clinical use.

Among several formulations proposed, drug encapsulation and nanomicelle-like formulation (based on mixtures of phospholipids and liquid triglycerides with or without cyclodextrins, able to self-assemble in an aqueous environment) have demonstrated the ability to increase fenretinide bioavailability at levels providing suitable plasma-tissue biodistribution and enhanced efficacy.

As reported in the previous sections, fenretinide nanomicelles, administered by intravenous injection at the dose of 30 mg/kg in animal models of neuroblastoma, provided high drug intratumor concentrations due to the nanomicelles’ ability to accumulate in solid tumors, by extravasation, through the discontinuities of the angiogenic tumor capillaries [48,49]. In brain tumor (DIPG-05 and RARE-08) animal models, the same intravenous doses of fenretinide nanomicelles increased overall survival, indicating their ability to cross the BBB [50,51]. By oral route, fenretinide nanomicelles showed significant efficacy at 100 mg/kg due to increased gastrointestinal fenretinide bioavailability that raised plasma levels and fenretinide concentrations in solid tumors [47]. Intraperitoneal administration of lower doses (10 mg/kg) of fenretinide nanomicelles extended survival and reduced neurological symptoms in an animal model of ALS [80].

Thus, according to the recent preclinical evidence, fenretinide nanomicelles can be regarded as antitumor nanomedicines at 30–100 mg/kg administration doses and, at lower doses, as potential therapeutic nanosystems in neurological diseases.

Recently, chemotherapeutic drugs such as kinase inhibitors, antimetabolites, alkylating agents and antibodies have been repurposed for neuroprotection, since it was demonstrated that some signaling pathways in neurodegeneration and cancer show peculiar sharing of genes and proteins that can become mutual therapeutic targets [8]. However, the major problems with the anticancer drugs used in neurodegenerative diseases are the presence of physical barriers in the brain, such as the blood–brain barrier, drug resistance and neuronal toxicity.

Drug resistance is mainly due to drug efflux transporters such as *p* glycoprotein (Pgp) and multidrug-resistant proteins (MRP), highly expressed at the blood–brain barrier, which limit the access of many drugs to the brain [109,110,111]. The independence of fenretinide from Pgp and MRP [112] and the ability of fenretinide to cross the blood–brain barrier, due to its highly lipophilic nature, make it a good candidate to treat different neurological diseases. Furthermore, appropriate formulations, such as nanomicelles, required to achieve adequate fenretinide bioavailability and biodistribution are already available to enhance its therapeutic response.

The neurotoxicity of many anticancer drugs [113], that strongly restricts their repositioning in neurological disease, was not observed with fenretinide in long-term administration studies as a chemopreventive agent [114] and in many other studies as a chemotherapeutic drug [34,115]. Additionally, the ability of fenretinide to decrease retinol plasma levels by binding RBP4 can reduce the incidence of ATRA-induced neurotoxicity observed in pediatric oncological patients [116], thus representing a safer alternative to ATRA in cancer.

Therefore, considering the amount of preclinical evidence, the low toxicity profile, the independence from drug efflux transporters, the ability to penetrate the brain, the newly available formulations and its hormetic behavior, fenretinide can be regarded as a potential drug candidate for selected neurological diseases, thus expanding its value beyond anticancer therapy.

## Figures and Tables

**Figure 1 ijms-23-07426-f001:**
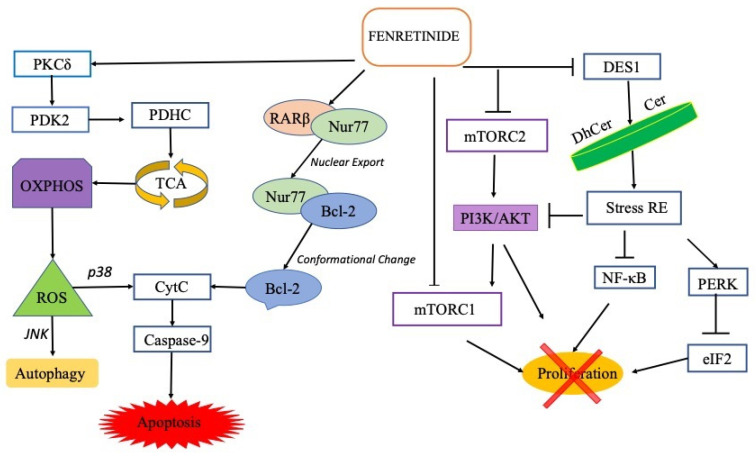
Schematic diagram of fenretinide activity in cancer. Pointed arrows represent pathway activation, and blunt arrows represent pathway inhibition. Fenretinide activates PKCδ in the mitochondrial signalosome with consequent activation of PDK2, PDHC, TCA and increased ROS production. High increase of ROS activates the *p*38 apoptotic pathway with CytC release, caspase-9 activation and induction of apoptosis. Moderate increase of ROS activates the JNK autophagy pathway.Fenretinide induces translocation of RARβ-Nur77 dimer from the nucleus to the cytoplasm where Nur77 binds Bcl-2 allowing their conformational change towards pro-apoptotic structures. Fenretinide inhibition of DES1 increases DhCer/Cer ratio and triggers endoplasmic reticulum stress with blocking of the PI3K/AKT/mTOR and NF-KB signaling pathways, as well as with blocking of eIF2 phosphorylation due to PERK. Moreover, fenretinide inhibits mTORC2 and mTORC1 by structural affinity with mTOR-ATP binding site.

**Figure 2 ijms-23-07426-f002:**
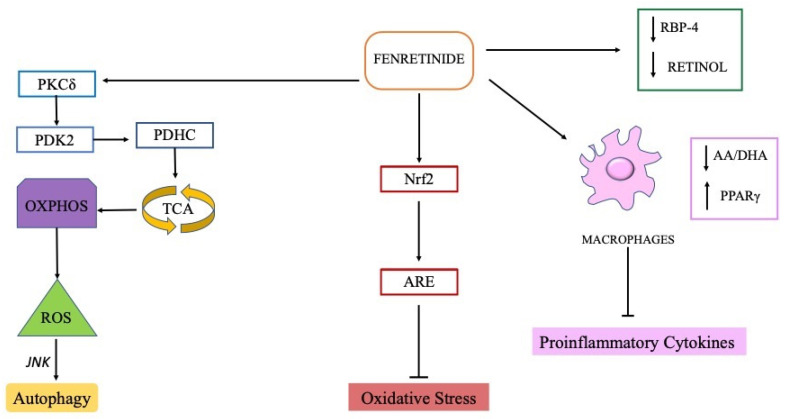
Schematic diagram of fenretinide activity in neurological diseases. Pointed arrows represent pathway activation, and blunt arrows represent pathway inhibition. Fenretinide activates PKCδ in the mitochondrial signalosome with consequent activation of PDK2, PDHC, TCA and increased ROS production. Low doses of fenretinide induce a moderate increase of intracellular ROS with activation of JNK autophagy pathway. Fenretinide decreases oxidative stress by improving the expression of the transcription factor Nrf2 that promotes the transcription of the antioxidant responsive element (ARE). Fenretinide downregulates the production of proinflammatory cytokines in macrophages by decreasing the AA/DHA ratio and enhancing the expression of PPARγ. Fenretinide binds RBP4 in place of retinol thereby decreasing circulating levels of both retinol and RBP4.

**Figure 3 ijms-23-07426-f003:**
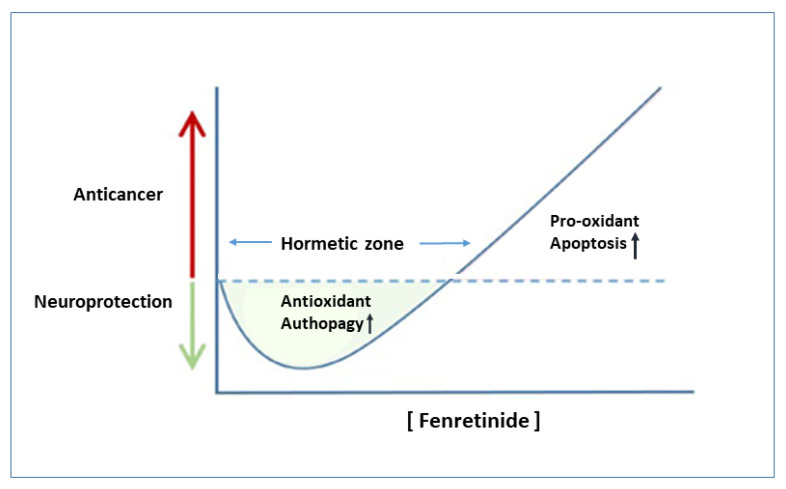
Hormetic dose response of fenretinide.

**Table 1 ijms-23-07426-t001:** Summary of studies evaluating the effects of fenretinide in cancer and neurological diseases. ↑ increased, ↓ decreased, bid *bis in die*.

Disease	Status of Investigation	Dosage and Formulation	Major Findings	References
**Neuroblastoma**	Phase I	100–4000 mg/m^2^ (soft gelatin capsules)	77% with stable dsease	[34,35]
		350–3300 mg/m^2^ bid (soft gelatin capsules)	1 complete response, 43% with stable disease	[36]
		352–2210 mg/m^2^ bid (LYM-X-SORB)	13% complete response, 20% with stable disease	[45]
**Renal cell cancer**	Phase II	900 mg/m^2^ bid (soft gelatin capsules)	37% with stable disease	[37]
**Lung cancer**	Preclinical *(xenograft mouse model)*	100 mg/kg (Bio-nFeR)	↓ tumour growth rate	[47]
	Phase II	900 mg/m^2^ bid (soft gelatin capsules)	30% with stable disease	[38]
**Prostate cancer**	Phase II	900 mg/m^2^ bid (soft gelatin capsules)	30% with stable disease	[39]
			56% with stable disease within 6 weeks	[40]
**Glioma**	Phase II	600–900 mg/m^2^ bid (soft gelatin capsules)	1 partial radiological response	[41]
**Ovarian Cancer**	Phase I/II	400–800 mg/bid (soft gelatin capsules)	no objective responses in advanced disease	[42]
	Phase II	900 mg/m^2^ bid (soft gelatin capsules)	42% stable disease	[43]
**Breast cancer**	Phase II	300–400 mg/d (soft gelatin capsules)	no objective responses in advanced disease	[44]
**Melanoma**	Preclinical (*xenograft mouse model*)	100 mg/kg (Bio-nFeR)	↓ tumour cell proliferation ↑ apoptosis	[47]
	Phase II	300–400 mg/d (soft gelatin capsules)	no objective responses in advanced disease	[44]
**Colon cancer**	Preclinical (*xenograft mouse model*)	150 mg/kg (Bio-nFeR)	↓ tumour cell proliferation ↑ apoptosis	[47]
**Advanced solid tumors**	Phase I	905–1414 mg/m^2^/die (lipid emulsion)	22% with stable disease	[46]
**Multiple sclerosis**	Preclinical (EAE mouse model)	3 mg/kg/die	↓ inflammation ↓demyelination ↓ neurological deficits	[69]
**Spinal cord injury**	Preclinical (contusion mouse model)	5 mg/kg/die	↓proinflammatory genes ↓oxidative stress ↓motor deficits	[77]
**Amyotrophic Lateral Sclerosis**	Preclinical (SOD1^G93A^ mice model)	5 mg/kg/die	↑ DHA/AA ratio ↓lipid peroxidation ↓gliosis ↓inflammatory mediators ↓motor deficit ↑female survival time	[78]
		10 mg /kg/die (NanoMFen)	↑female survival time	[79]
**Alzheimer disease**	Preclinical (BACE1 mice model)	40 mg/kg/die	↓ serum leptin ↓ adiposity gain	[87]
			↓ neuroinflammation ↓ ER stress ↑ sAPPα ↓ Aβ ↑ memory deficit	[88]
**Depression**	Preclinical (LPS mice model)	20–40 mg/kg	↑ Nrf2 signalling ↑ BBB dysfunction ↓ depressive behaviour	[66]
**Macular degeneration**	Phase I	100–300 mg/die	↓ serum RBP-retinol ↓ lesion growth rates	[102]

## Data Availability

Not applicable.

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
