# Peer review of "Fenretinide in Cancer and Neurological Disease: A Two-Face Janus Molecule"

_ijms, 2022, doi:10.3390/ijms23137426_

Round 1

Reviewer 1 Report

Thanks to the authors for their work. But a substantial revision of the text is needed for this level of magazine. At the moment, the text is not scientific enough. It is necessary to shift the emphasis, to strengthen the text. At least two people basically wrote this article. One of them is not competent. Please rewrite. 

1. Introduction. There is no information about what methods of medical treatment are currently used for the diseases described. There is no comparison of why some drugs are worse and some worse. It is necessary to convince the reader that Fenretinide has great potential. This is only possible by comparison. he text makes very general sense. No specifics. It is not correct.

2. Figure 1 is as general as the rest of the introduction. You need to be specific about what you want to show in the figure.

3.in the description of ROS, there is no information on how the action of Fenretinide relates to the intrinsic ways of fighting ROS (catalase-superoxide dismutase cascades, etc.)

4.  A promising sentences that ends with nothing. Rewrite, expand, and talk about this research right here. "To date 40 clinical trials have been reported in ClinicalTrials.gov registry testing fenretinide in diverse types of cancer including prostate, breast and neuroblastoma."

5. Increased fenretinide plasma concentrations are expected to raise the drug levels in the tumor mass allowing the achievement of suitable drug concentrations for activity. - for other substances it is not so? is this a new conclusion?

6. The in vivo studies, carried out on fenretinide until now, reported high variability of 187 results indicating that the minimal active concentration is not always achieved into the 188 tumor mass. This depends on the tumor permeability towards drug penetration and the 189 drug plasma concentration that regulate the drug distribution between tumor and blood. - there are no words! is this a new conclusion for this topic? really? is this something that needs to be reported in this review and in this journal?

7.The unpublished doctoral thesis of Skinner T (78), an author of the paper of Lopez- 373 Vales et al (77), describes a beneficial role of the same doses of fenretinide in a faster pro- 374 gressive mice model of amyotrophic lateral sclerosis (ALS), a fatal neurodegenerative dis- 375 ease characterized by selective death of motor neurons causing progressive muscle atro- 376 phy and spasticity. - is it possible to refer to unpublished material?

the article needs to be refined. It should not be printed in this form.

Author Response

Reviewer Comment 1 (RC.1) Introduction. There is no information about what methods of medical treatment are currently used for the diseases described. There is no comparison of why some drugs are worse and some worse. It is necessary to convince the reader that Fenretinide has great potential. This is only possible by comparison. he text makes very general sense. No specifics. It is not correct.. 

Author’s Reply Introduction has been modified: the main classes of chemotherapy drugs used in neurodegenerative diseases have been reported with their main drawbacks in comparison with Fenretinide.

RC.2  Figure 1 is as general as the rest of the introduction. You need to be specific about what you want to show in the figure.

Author’s Reply We agree with the comment thus, Figure 1 has been deleted

RC.3 In the description of ROS, there is no information on how the action of Fenretinide relates to the intrinsic ways of fighting ROS (catalase-superoxide dismutase cascades, etc.)

Author’s Reply We have reported that one of the fenretinide mechanisms in inducing cell death or autophagy is ROS over production due to its ability to replace retinol in the mitochondrial signalosome lines (259-269).

RC.4 A promising sentences that ends with nothing. Rewrite, expand, and talk about this research right here. "To date 40 clinical trials have been reported in ClinicalTrials.gov registry testing fenretinide in diverse types of cancer including prostate, breast and neuroblastoma."

Author’s Reply We have modified according to the reviewer’s suggestions.

RC.5 Increased fenretinide plasma concentrations are expected to raise the drug levels in the tumor mass allowing the achievement of suitable drug concentrations for activity. - for other substances it is not so? is this a new conclusion?

Author’s Reply In order to take into account the referee’s comment, in the revised version we have clarified that increasing plasma concentrations at levels suitable to obtain active drug concentrations in the tumor mass is difficult for drugs characterized by low aqueous solubility such as Fenretinide.

RC.6 The in vivo studies, carried out on fenretinide until now, reported high variability of 187 results indicating that the minimal active concentration is not always achieved into the 188 tumor mass. This depends on the tumor permeability towards drug penetration and the 189 drug plasma concentration that regulate the drug distribution between tumor and blood. - there are no words! is this a new conclusion for this topic? really? is this something that needs to be reported in this review and in this journal?

Author’s Reply In the revised version, we have rephrased in a more general way.

RC.7 The unpublished doctoral thesis of Skinner T (78), an author of the paper of Lopez- 373 Vales et al (77), describes a beneficial role of the same doses of fenretinide in a faster pro- 374 gressive mice model of amyotrophic lateral sclerosis (ALS), a fatal neurodegenerative dis- 375 ease characterized by selective death of motor neurons causing progressive muscle atro- 376 phy and spasticity. - is it possible to refer to unpublished material?

Author’s Reply In our last paper (Orienti et al 2021) we already cited the seminal work of Skinner (available in the web) as specifically requested by a referee ; furthermore, in the IJMS Instructions for Authors it is clearly described how to cite a thesis work, suggesting the will of the journal to include these writings in the references of their publications.

RC.8 the article needs to be refined. It should not be printed in this form.

Author’s Reply We have refined according to the reviewer’s suggestions, hoping it is now suitable for publication.

Reviewer 2 Report

The article describes in detail the activities of fenretinide in neoplastic and neurologic pathologies and the possible mechanisms of action underlying these applications.

Overall, the article is a source of information of potential interest to readers interested in various fields and specialties. Overall, the work is well organized and documented, but there are some aspects that can be revised and improved.

- Lines 31-33. Review and correct: decreases in anti-apoptotic proteins and increases in pro-apoptotic molecules favor apoptosis and not its inhibition.

- The concept described in lines 67-71 is not very clear, please rephrase the sentence.

- The concepts described in lines 170-179 have already been described, this part is redundant.

- Lines 295-298. Does fenretinide enhance PPARgamma expression or ligands of this receptor or both? Not clear.

- Lines 312-320. The concepts presented have already been described, this part is redundant.

- Because the review is very rich in information about different pathologies and refers to studies in vitro, in vivo in experimental preclinical models, and clinical trials, a table or figure showing the different status of knowledge and application of fenretinide would be useful to give the reader an overview at a glance.

-The English language should be checked for some errors.

- Some abbreviations are not defined (e.g., MBP line 331).

Author Response

RC.1 - Lines 31-33. Review and correct: decreases in anti-apoptotic proteins and increases in pro-apoptotic molecules favor apoptosis and not its inhibition.

Author’s Reply  We thank the referee for his/her positive comments. We apologize for the mistake; the text has been corrected in the revised version

RC.2 The concept described in lines 67-71 is not very clear, please rephrase the sentence.

Author’s Reply The concept described in lines 67-71 has been rephrased.

RC.3 The concepts described in lines 170-179 have already been described, this part is redundant.

Author’s Reply The concepts described in lines 170-179 belong to the Caption of Figure 2 (now Figure 1) but it was incorrectly included as text. We have now corrected the layout and marked the captions of the figures in italics.

RC.4 - Lines 295-298. Does fenretinide enhance PPARgamma expression or ligands of this receptor or both? Not clear.

Author’s Reply The concept in lines 295-298 has been now clarified in lines 543-547

RC.5 Lines 312-320. The concepts presented have already been described, this part is redundant

Author’s Reply The concepts described in lines 312-320 belong to the Caption of Figure 3 (now Figure 2) but, it was incorrectly included as text. We have now corrected the layout and marked the captions of the figures in italics.

RC.6- Because the review is very rich in information about different pathologies and refers to studies in vitro, in vivo in experimental preclinical models, and clinical trials, a table or figure showing the different status of knowledge and application of fenretinide would be useful to give the reader an overview at a glance.

Author’s Reply According to the reviewer’s suggestion, a table (Table 1. Summary of studies evaluating the effects of fenretinide in cancer and neurological diseases) has been added in the revised version

RC.7 The English language should be checked for some errors.

Author’s Reply We have checked and corrected English throughout the manuscript

RC.8 Some abbreviations are not defined (e.g., MBP line 331).

Author’s Reply We apologize for the forgetfulness; now all the abbreviations have been defined in the text.

Reviewer 3 Report

I have found the review by Ponteza, Lodesertp and Orienti very interesting and detailed. They overview the underlying mechanism of fenretinide on many different paths and how they confluent on the treatment of very opposite diseases involving cancer or neurodegeneration. Instead of giving a detailed explanation of the effect of the drug in each path, they provide an overview of these combined with the importance of the development of novel formulations and administration ways to improve efficiency on cancer treatment. They also explain the routes to use this drug in neurodegenerative diseases concluding how a single drug can be used for one or another application depending on the dose and usage.

I only found minor issues with figures and found some edition issues in the manuscript. I’m not an English native speaker and I do not feel confident for editing but I think some parts of the manuscript will benefit from a native English editing.

Figures: in general, I have found that figures captations are too brief and reader will benefit from a little bit of explanation.

Figure 1: Structure of fenretinide should be bigger and arrows showing if there is an increase or a decrease of the corresponding protein/route (RAR, DES-1; RBP-4 and ROS) will be self-explanatory. Considering the text, the figure is not entirely correct at the Cer/DhCer part (there is an unbalance of these proteins but it is because an increase of DhCer, while Cer is not affected by fenretinide). On the other hand, I have found another miss match between the figure and the text. Figure is referred in the section of ‘Fenretinide in cancer’, where it affects at least in 4 different mechanisms: RAR, DES1, MTOR and ROS (mitochondrial signalosome). In the figure appears RBP4 (instead of MTOR), which is mentioned before explaining the interaction of the drug with this protein.

Figure 2: There are proteins that are not mention in the text and presented in the figure (like caspase-9) and others that I do not know what are they (TCA, OXPHOS). In addition, the levels of DhCer and Cer should be the other way around because fenretinide increase DhCer, not Cer

Figure 3: same issues that figure 2 related to TCA and OXPHOS

Edition: Most of the review is find but sometimes they include the acronym of a protein or pathway and do not mention where this come from. For instance, TLRs and TLRs mediated pathways in line 290, MBP in line 331, i.p route in line 387 or iv route in line 513.

They often include the name of protein/pathway and then mention the acronym in parenthesis, but sometimes they do it the other way around (for example PDK2 in line129)

Author Response

RC1 I only found minor issues with figures and found some edition issues in the manuscript. I’m not an English native speaker and I do not feel confident for editing but I think some parts of the manuscript will benefit from a native English editing.

Author’s Reply We have edited English throughout the manuscript

RC.2 Figures: in general, I have found that figures captations are too brief and reader will benefit from a little bit of explanation

Author’s Reply Comprehensive captions were already present but were incorrectly included as text. We have now marked the captions of the figures in italics

RC.3 Figure 1: Structure of fenretinide should be bigger and arrows showing if there is an increase or a decrease of the corresponding protein/route (RAR, DES-1; RBP-4 and ROS) will be self-explanatory. Considering the text, the figure is not entirely correct at the Cer/DhCer part (there is an unbalance of these proteins but it is because an increase of DhCer, while Cer is not affected by fenretinide). On the other hand, I have found another miss match between the figure and the text. Figure is referred in the section of ‘Fenretinide in cancer’, where it affects at least in 4 different mechanisms: RAR, DES1, MTOR and ROS (mitochondrial signalosome). In the figure appears RBP4 (instead of MTOR), which is mentioned before explaining the interaction of the drug with this protein.

Author’s Reply Figure 1 has been deleted since, according to the reviewer 1, it generated confusion whereas the mechanisms represented in Figure 1 are more clearly reported in Figure 2 and 3 (now 1 and 2).

RC.4 Figure 2: There are proteins that are not mention in the text and presented in the figure (like caspase-9) and others that I do not know what are they (TCA, OXPHOS). In addition, the levels of DhCer and Cer should be the other way around because fenretinide increase DhCer, not Cer

Author’s Reply Figure 2 (now Figure 1) has been improved. Caspase-9, TCA and OXPHOS have been mentioned in the text. In Figure 2 (now Figure 1) DhCer and Cer levels are represented by a green disk bent towards the dhCer to simulate the plate of a scale bent toward the heavier component. This is a graphical representation sometimes used in the literature.

RC.5 Figure 3: same issues that figure 2 related to TCA and OXPHOS

Author’s Reply TCA and OXPHOS have been mentioned in the text

RC.6 Edition: Most of the review is find but sometimes they include the acronym of a protein or pathway and do not mention where this come from. For instance, TLRs and TLRs mediated pathways in line 290, MBP in line 331, i.p route in line 387 or iv route in line 513.

They often include the name of protein/pathway and then mention the acronym in parenthesis, but sometimes they do it the other way around (for example PDK2 in line129)

Author’s Reply We apologize for the forgetfulness; all the acronyms have been now defined in the text